# Development of Alternative Method for Manufacturing Structural Zirconium Elements for Nuclear Engineering

**DOI:** 10.3390/ma14175006

**Published:** 2021-09-02

**Authors:** Kirill Ozhmegov, Anna Kawalek, Dariusz Garbiec, Henryk Dyja, Alexandr Arbuz

**Affiliations:** 1Faculty of Production Engineering and Materials Technology, Czestochowa University of Technology, 69 J.H. Dabrowskiego St., 42-201 Czestochowa, Poland; kvozhmegov@wp.pl (K.O.); kawalek.anna@pcz.pl (A.K.); 2Łukasiewicz Research Network – Metal Forming Institute, 14 Jana Pawla II St., 61-139 Poznan, Poland; henryk.dyja48@gmail.com; 3Core Facilities, AEO Nazarbayev University, 53 Kabanbay Batyr Ave., Nur-Sultan 010000, Kazakhstan

**Keywords:** zirconium, powder metallurgy, spark plasma sintering, microstructure, rheological, dilatometric studies

## Abstract

Zirconium is used as a structural material for use in aggressive environments, including the core of nuclear reactors. The traditional technology of manufacturing the structural elements of zirconium nuclear reactors is characterized by a long technological process and a significant amount of waste in the form of metal shavings. The paper presents the results of an alternative technology, spark plasma sintering, for manufacturing zirconium products. A complex of microstructural and mechanical studies of the obtained samples was carried out according to the ASTMB-351 standard. The sintering of zirconium powder and options for subsequent processing by various methods, including non-standard ones such as radial shear rolling, are justified.

## 1. Introduction

Due to its unique combination of physico-mechanical and thermo-physical properties, as well as its low thermal neutron capture cross-section (0.18 ± 0.02 barn; 1 b = 10^−28^ m^2^), zirconium is widely used as a structural element for use in the nuclear power industry [1,2]. Zirconium is also characterized by exceptionally high corrosion resistance with sufficiently good mechanical properties in aggressive environments; therefore, semi-finished and finished products made of zirconium are manufactured for chemical equipment, as well as elements of electrical and electronic devices (electronic lamps, capacitors, rectifiers) [3]. Zirconium products are also used in surgery, since they do not cause undesirable consequences in the human body [4].

The traditional technological route of manufacturing zirconium products consists of the following operations [2,3,5]:Briquetting by pressing and sintering zirconium powder;Ingot smelting;The production of workpieces employing hot pressure treatment methods (forging, rolling) and mechanical processing;Hot pressing;The production of finished products by cold pressure treatment methods (rolling, drawing) with intermediate thermal, mechanical, and chemical treatments.

The traditional technology of manufacturing zirconium products is labor-intensive and costly due to the significant number of pressure treatment processes and intermediate technological operations of thermal, chemical, and mechanical treatments. The efficiency of these processes is low due to the large technological waste from individual technological operations [2]. However, the process can be improved and shortened. The use of screw rolling mills for solid round products is an effective way from the point of view of reducing material and time costs and creating a fine-grained uniform microstructure [6]. After hot pressing, the rod is compressed to the finished size under conditions of cold or hot deformation, while the number of cycles of deformation and thermal operations is reduced compared to traditional technology.

The article presents a new method for manufacturing zirconium semi-finished products by the spark plasma sintering (SPS) of zirconium powder. In recent years, the spark plasma sintering technique has been increasingly used for the consolidation of powder materials such as composites, metals, carbides, nitrides, borides, fluorides, and others. Spark plasma sintered compacts are usually characterized by better physical and mechanical properties compared to products manufactured by traditional techniques. The materials obtained by spark plasma sintering are also characterized by a uniform and homogeneous microstructure over the entire volume of the sintered compacts [7]. In this technique, periodically repeated direct current (DC) pulses are used to directly heat the compacted powder. Sintering is shorter (several minutes), and it is carried out at a lower temperature. Rapid heating and cooling, as well as the short processing time, do not allow significant grain growth.

The article presents the new technology route of manufacturing semi-finished products from zirconium powder in one operation by spark plasma sintering. The semi-finished products have a homogeneous recrystallized microstructure and strength properties in accordance with the requirements the ASTM B-351 “Specification for hot-rolled and cold-finished zirconium and zirconium alloy bars, rod, and wire for nuclear application” standard [8].

## 2. Materials and Methods

For the investigations, zirconium powder was used with the chemical composition shown in Table 1 and fractional composition of the powder particles shown in Figure 1. Zirconium powder was obtained by the electrolysis of molten zirconium salts [2,3].

The zirconium powder was sintered using an HP D 25/3 SPS furnace (FCT Systeme, Rauenstein, Germany). The sintering was carried out using tools made from 2333 grade graphite (Mersen, Gennevilliers, France). Copper or tungsten foil was placed between the powder, die, and punches, and then graphite foil, as presented schematically in our patent [9]. Copper or tungsten foil was used to protect the zirconium from reaction with the carbon of the graphite tools, which negatively affects the corrosion resistance [10]. The choice of foil was carried out in relation to the sintering temperature and the melting temperature of these materials. Therefore, at the higher temperature (>1000 °C), tungsten foil was used, and at the lower temperatures (<1000 °C), copper foil was used. The sintering process was carried out in vacuum of 5 × 10^−2^ mbar at temperatures of 700, 750, 800, and 1400 °C at a heating rate of 100 °C/min and compacting pressure of 80 MPa. The holding time was 10 min. The cooling rates were 400 °C/min and 10 °C/min. Cylindrical discs with dimensions of Ø40 × 12 mm, Ø30 × 11 mm, and Ø20 × 30 mm were obtained.

After sintering at the temperature of 800 °C, the zirconium samples were heat treated at a temperature of 580 or 750 °C, for a holding time of 120 or 30 min respectively, followed by slow cooling to ambient temperature.

The scheme of sample preparation for assessing the effect of the cooling rate after sintering and subsequent heat treatment on the formation of the microstructure and properties is presented in Table 2.

Density measurements were carried out by the Archimedes method in accordance with the ISO 3369:2006 standard [11] using an EX225DM (Ohaus, Parsippany, NJ, USA) scale. Microhardness measurements were conducted by the Vickers method in accordance with the ISO 6507-1:2007 standard [12] using an FM-700 (Future-Tech, Kawasaki, Japan) hardness tester at a load of 0.4903 N, and the holding time was 15 s. Microstructure observations were performed using an Eclipse L150 (Nikon, Kawasaki, Japan) light microscope (LM) and a Crossbeam 540 (Carl Zeiss, Oberkochen, Germany) scanning electron microscope (SEM). The phase composition was analyzed by backscattered electron diffraction (EBSD) using a NordlysNano detector (Oxford Instruments, Abingdon, UK) and AZtec, Channel5, and Tango software (Oxford Instruments). The specimens for analysis were prepared by electrolytic etching on a LectroPol-5 (Struers, Copenhagen, Denmark) unit at the voltage of 20.5 V/40 s in a special Struers A2 electrolyte at the temperature of −20 °C. The specimens were pre-mechanically ground on a Sapphire 520 (ATM, Mammelzen, Germany) grinding and polishing machine. The mechanical properties were investigated by the compression test using a Gleeble 3800 (Dynamic Systems, Poestenkill, NY, USA) plastometer in accordance with the GOST 25.503-97 technical standard [13]. In order to assess the microstructural transformations, as well as the rheological behavior of the obtained materials, plastometric and dilatometric studies using a Gleeble 3800 plastometer and a DIL805 A/D (TA Instruments, New Castle, DE, USA) dilatometer were carried out. The methodology of these studies is presented in [14,15,16]. The analysis of the research results was conducted in accordance with ASTM B-351 [8] as well as experimental data of zirconium products manufactured using traditional technology [4,6,10,17].

## 3. Results and Discussion

### 3.1. Investigation of Effect of Spark Plasma Sintering Regime on Density and Microhardness

A photograph of a spark plasma sintered zirconium sample Ø20 × 30 mm is shown in Figure 2.

The sintering curves are presented in Figure 3. The main difference between the two sintering regimes is the cooling rate—rapid at 400 °C/min and slow at 10 °C/min, presented in Figure 3a,b, respectively. What is clearly seen from the punch displacement curves is that the 10 min holding time allows compacts with nearly full density to be obtained, having a relative density of 98.31% and 98.92% for the rapid and slow cooling, respectively. The plateau of both curves is clearly seen during the holding stage; however, at the end of sintering, a slight punch movement was observed, which means that further consolidation (residual porosity elimination) occurred.

Based on the data presented in Table 3, it can be concluded that the optimal sintering temperature of zirconium powder is 800 °C because the samples obtained at this temperature are characterized by the highest microhardness, and their bulk density is almost full (zirconium density is 6.51 g/cm^3^). In view of the fact that the bulk density and microhardness of the samples sintered at temperatures lower than 800 °C were not sufficient, these samples were not studied further.

Although similar results were obtained after sintering zirconium powder at the temperature of 1400 °C, sintering at such a high temperature leads to increased costs (for example, increased energy consumption, and it is not possible to use the cheaper copper foil). After sintering, the samples were cooled utilizing various rates—400 °C/min and 10 °C/min. The use of slow cooling after zirconium powder sintering increases the time of the process, but it allows a more pronounced recrystallized microstructure to be obtained.

### 3.2. Investigation of Effect of Spark Plasma Sintering Cooling Rate and Subsequent Heat Treatment Regime on Microstructure and Microhardness

As part of the study, tests were conducted to determine the sintering regimes and the regimes of subsequent heat treatment to meet the requirements of the ASTM B-351 standard [8]. The requirements specify that zirconium products must have at least a 90% recrystallized microstructure.

Sintering at 1400 °C followed by cooling at 400 °C/min leads to the formation of a martensitic microstructure (Figure 4). Such a microstructure is formed during rapid cooling of the initial β-phase. A nonequilibrium α′-phase is formed in the metal, which is undesirable [18].

A decrease in the sintering temperature to 800 °C with subsequent cooling at 400 °C/min and 10 °C/min leads to a decrease in the nonequilibrium α′-phase. Moreover, at the cooling rate of 10 °C/min, a more noticeable transition of the metal microstructure to the equilibrium state is observed (Figure 5a,b). The use of cooling at lower rates <10 °C/min is impractical for the spark plasma sintering technique.

Another way to meet the requirements of the ASTM B-351 standard [8] in terms of microstructure is to conduct heat treatment after spark plasma sintering. To determine the effect of the heat treatment regime, the samples after sintering at 800 °C and cooling at 400 °C/min were heated to 580 °C and 750 °C using a vacuum furnace built into a Gleeble 3800 plastometer with a residual pressure not higher than 10^−4^ torr. In this temperature range, recrystallization takes place in metals and alloys based on zirconium [19,20,21].

Figure 5c,d shows the microstructure of zirconium specimens obtained as a result of spark plasma sintering followed by vacuum heat treatment at 580 °C at the holding time of 120 min and 750 °C at the holding time of 30 min.

As a result of vacuum annealing, an equilibrium microstructural state of the specimens is formed. The microstructure is recrystallized with a grain size of 5–30 µm. At the same time, there are single large grains up to ≈30 × 80 µm in size. Moreover, at the heat treatment temperature of 750 °C, the number of these large grains in the specimens is higher than after treatment at the temperature of 580 °C. To reduce the average size, it is advisable to consider reducing the holding time. Thus, the best result is achieved with heat treatment at 580 °C. This sample, as the most successful, was studied in more detail by SEM and EBSD. The advantage of EBSD over other methods of phase analysis (for example XRD) is that it is possible to visually identify the phases of interest. Figure 6a shows the phase composition of the specimen after sintering at 800 °C and subsequent heat treatment at 580 °C. The map step was 0.83 µm, and the map size was 544 × 408 points.

As can be seen from Figure 6, an extra-small amount of residual beta phase (at least 1%) is still observed in the specimen consisting of an equilibrium α-matrix (green). The remnants of the β-phase (purple) are observed mainly along the grain boundaries, and in some cases, in the form of inclusions. They most likely appeared along the most prominent boundaries of the powder particles at the time of current transmission and plasma formation. The inclusions probably represent a part of the boundary of the upper grain (ground during specimen preparation). An unambiguous answer to this question would be given by the use of 3D EBSD [22]. The IPF map shows an equiaxial microstructure with predominantly large-angle inter-grain boundaries identical to the one shown in Figure 5c. The microstructure has no pronounced texture and should provide the finished product with isotropic properties. When analyzing a sintered specimen, an important characteristic is the absence of pores.

Metallographic analysis of the specimens did not reveal pore chains, which is typical for the formation of the microstructure in the traditional way, when a deformation porosity is formed in the axial part of the products. At the same time, there are single pores up to 3–4 µm in size.

The microhardness after vacuum heat treatment at the temperature of 580 °C is 150 ± 10 HV_0.05_, and at the temperature of 750 °C, it is 160 ± 10 HV_0.05_ and corresponds to the microhardness values of the finished material from the Zr-1Nb alloy obtained by traditional technology [4,10]. What is worth noting is the significantly higher hardness reported by Jaworska et al. [23], where the Zr-1Nb alloy was manufactured by spark plasma sintering. The hardness is >339 HV_2_ after spark plasma sintering at 1200 °C and 1 min holding time and depends on the used niobium powder. If the powder is finer, the hardness is slightly lower.

### 3.3. Dilatometric Tests of Spark Plasma Sintered Zirconium

To determine the kinetics of the microstructural transformations in zirconium samples during heat treatment, dilatometric studies were performed. The heating and cooling rates are shown in Table 2. Figure 7 shows the dilatometric curves obtained by heat treatment of zirconium spark plasma sintered at 800 °C and subsequent cooling at 400 °C/min.

It can be seen from the presented curves that thermal expansion of the metal occurs during heating; as a result, an increase in the size of the specimens is observed on the dilatometric curves. Moreover, the nature of the change in the curves up to 580 °C is approximately the same. When entering the 580 °C regime at the holding time of 120 min (Figure 7a), the metal continues to expand further to ≈100 µm. The expansion is most probably associated with a decrease in the nonequilibrium α′-phase and the transition of the metal to an equilibrium state during recrystallization.

From Figure 7b, it can be concluded that with an increase in the heating temperature from 580 to 750 °C, the expansion of the metal continues to ≈100 µm, but when entering the annealing, no significant changes are observed on the dilatometric curve. It is most likely that significant changes in the microstructure began at a lower temperature of ≈500 °C; a change in the linear expansion coefficient (1) is noted on the curve.
(1)$$\alpha_{L}\;=\;\frac{1}{L}{\left(\frac{\partial L}{\partial T}\right)}_{p},\;K^{-1}$$

When the metal is cooled down, the dilatometric curves descend to the region of negative values. This means that the value of thermal compression is slightly greater than the value of thermal expansion. This very likely indicates different values of linear expansion coefficient *αL* of the metal in the nonequilibrium and equilibrium states. The difference in the values on the dilatometric curves after cooling at ambient temperature may be due to the tolerances for geometric dimensions during the manufacture of samples.

The results presented in Figure 7 confirm the metallographic studies carried out in terms of the feasibility of heat treatment to create an equilibrium microstructure after spark plasma sintering zirconium powder. Moreover, the annealing regime must be chosen to ensure the recrystallization state. The annealing regimes proposed in the work provide this state. At the same time, it follows from the comparison of the diagrams that heat treatment at 580 °C and holding at 120 min is sufficient to exclude the nonequilibrium α′-phase.

### 3.4. Results of Mechanical and Rheological Tests of Spark Plasma Sintered Zirconium. Comparison of Results with Traditional Manufacturing Route

The mechanical properties were investigated by means of the compression test on a Gleeble 3800 plastometer of cylindrical specimens having the dimensions Ø10 × 12 mm after heat treatment at 580 °C and holding at 120 min. The specimens were cut by wire electrical discharge machining from spark plasma sintered samples with the dimensions Ø40 × 12 mm.

Analysis of the results of the mechanical compression tests at the temperature of 20 °C and the strain rate of 10^−3^ s^−1^ showed that the value of the yield strength and tensile strength of the specimens obtained using the new technology route meet the requirements of the ASTM B-351 standard [8] (Table 4).

Rheological tests were carried out in relation to the conditions of cold pressure treatment in the range of strain rates from 0.1 to 5.0 s^−1^ on the Gleeble 3800 plastometer using the compression test of cylindrical specimens Ø10 × 12 mm after heat treatment at 580 °C and holding at 120 min. The deformation speed range of cold pressure treatment is interesting, primarily in terms of determining the manufacturability under significant loads on the material.

Figure 8 shows the results of comparative rheological studies of samples made of zirconium alloy obtained using the new technology and Zr-1Nb made by means of traditional technology using rolling on an RSR mill [4]. The comparison of alloys similar in chemical composition was carried out in order to show the possibility of interchanging these materials. More extensive data on the rheological properties of the Zr-1Nb alloy are presented in [15,17].

From the comparison of the flow curves of the investigated specimens obtained using the new (Zr) and traditional (Zr-1Nb) manufacturing routes, it was revealed that the flow curves for the specimens manufactured using the new route are slightly lower than the curves for the specimens obtained using the traditional technology (Figure 8), which is due to the difference in chemical composition. The samples obtained using traditional technology have a niobium addition (≈1 wt %). Niobium is an alloying element introduced into the chemical composition of ingots to stabilize the corrosion resistance of unalloyed zirconium (it eliminates the harmful effect of small amounts of carbon, aluminum, and titanium impurities) and improves the mechanical properties. Niobium effectively reduces the fraction of hydrogen absorbed by zirconium alloys [2].

The ambiguous effect of increasing the strain rate on the deformation resistance of the metal should also be noted. The value of the deformation resistance for the specimens obtained by the traditional route increases by about 10% with an increase in the strain rate from 0.1 to 5.0 s^−1^. At the same time, the thermal effect of plastic deformation ∆T reaches 140 °C, at strain ε = 0.4. On the other hand, an increase in the strain rate when testing the specimens made using the new technology has the opposite effect. At deformation rate ε = 0.1 s^−1^, the value of plasticizing stress increases in the entire examined range of deformations; nonetheless, after exceeding deformation ε = 0.4, the increase becomes less pronounced. Such a course of the strengthening curve is related to the increasing influence of the thermal plastic deformation effect. The flow curve at the strain rate of 5.0 s^−1^ passes below the flow curve at the strain rate of 0.1 s^−1^. Moreover, before strain ε = 0.4, the difference between the curves first increases, and then it decreases with rising strain. This phenomenon can be explained by the greater sensitivity of the metal to an increase in temperature during deformation. When the strain is up to ε = 1.1 and the strain rate is 5.0 s^−1^, the thermal effect of plastic deformation for specimens made using the new technology is ≈200 °C.

The value of the maximum strain level ${Ʌ}_{p}\;=\;\sqrt{3}\ln\left(\frac{h_{0}}{h_{p}}\right)$ at ${\left(\frac{\sigma_{cp}}{\tau_{i}}\right)}_{cp}\;=\;-0.5$ [24] (where *h_p_* is the height of the deformable specimen at the moment of formation of a crack in the metal) is significantly higher for the specimens obtained using the new technology. Thus, for the specimens obtained by the new technology, the value of the maximum strain level is Λ*_p_* = 1.7, instead of Λ*_p_* = 0.6 when employing the traditional technology. This indicates that the metal is highly technologically advanced with sufficient strength. Due to the negative radiation effect on the plasticity of products made of zirconium alloys, an additional resource of plasticity can extend the service life of the product in the reactor core.

## 4. Conclusions

Analysis of the studied samples (Ø20 × 30 mm, Ø30 × 11 mm and Ø40 × 12 mm) revealed that spark plasma sintering provides materials with nearly full density. Subsequent heat treatment at the temperature of 580 °C allows a recrystallized microstructural state to be obtained, with a grain size of 5–30 µm and single large grains up to ≈30 × 80 µm in size. No microstructural or phase inhomogeneities were detected in the microstructure of the specimens. The microhardness of the zirconium obtained by the new technology is about 40% higher than the material made by the traditional route described in the introduction. The increased microhardness may make it relevant to use such special methods as radial-shear rolling for further processing.

The analysis of the research results demonstrated that the samples made by spark plasma sintering at the temperature of 800 °C and subsequent heat treatment meet the requirements for the microstructural state as well as the mechanical property requirements regarding the yield strength and the tensile strength in the ASTM B-351 standard. Thus, the zirconium obtained by the new route can be considered to be used as structural elements, including the material of fuel element shell plugs.

The results of additional comparative tests of the zirconium showed similar values in microhardness and deformation resistance to the Zr-1Nb alloy. At the same time, the material made by spark plasma sintering is characterized by higher plasticity. An increase in plasticity while maintaining the strength characteristics of the material can extend the service life of products.

The new spark plasma sintering technology of manufacturing zirconium, e.g., of bars for fuel element shell plugs in the future can eliminate the long traditional bar manufacturing cycle. In addition, it is possible to significantly raise the yield of suitable finished products due to the absence of waste when trimming in the individual processing stages.

The use of finished products obtained by the new technology in the nuclear power industry requires further testing under conditions similar to those that exist in the core of a nuclear reactor. This section is not mandatory but can be added to the manuscript if the discussion is unusually long or complex.

## 5. Patents

The research results presented in this manuscript are protected by intellectual property in the Republic of Poland on the basis of patent application No. 427642.

## Figures and Tables

**Figure 1 materials-14-05006-f001:**
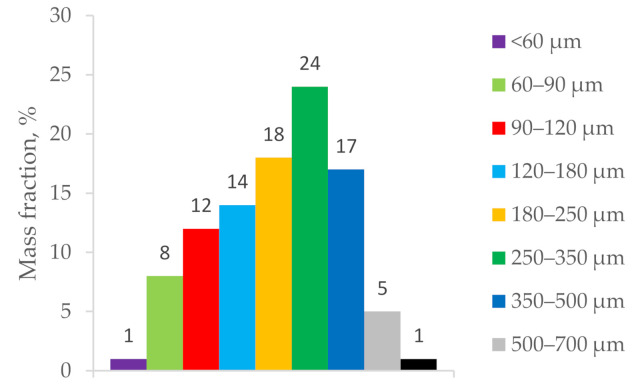
Fractional composition of zirconium powder.

**Figure 2 materials-14-05006-f002:**
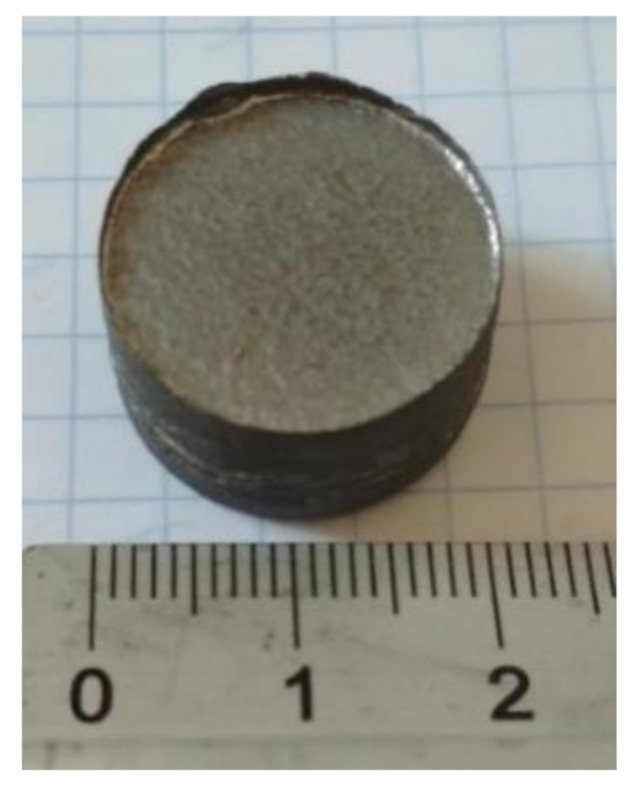
Photograph of spark plasma sintered zirconium sample with dimensions Ø20 × 30 mm.

**Figure 3 materials-14-05006-f003:**
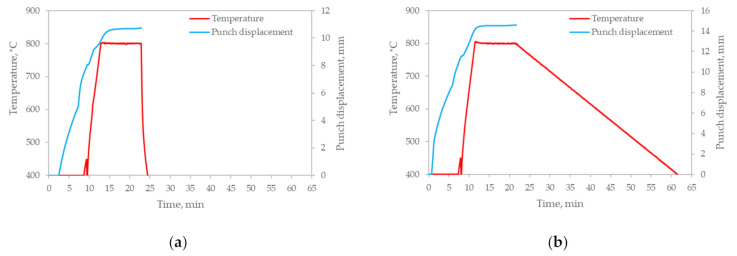
Sintering curves of zirconium powder spark plasma sintered at 800 °C at various cooling rates: (**a**) 400 °C/min; (**b**) 10 °C/min.

**Figure 4 materials-14-05006-f004:**
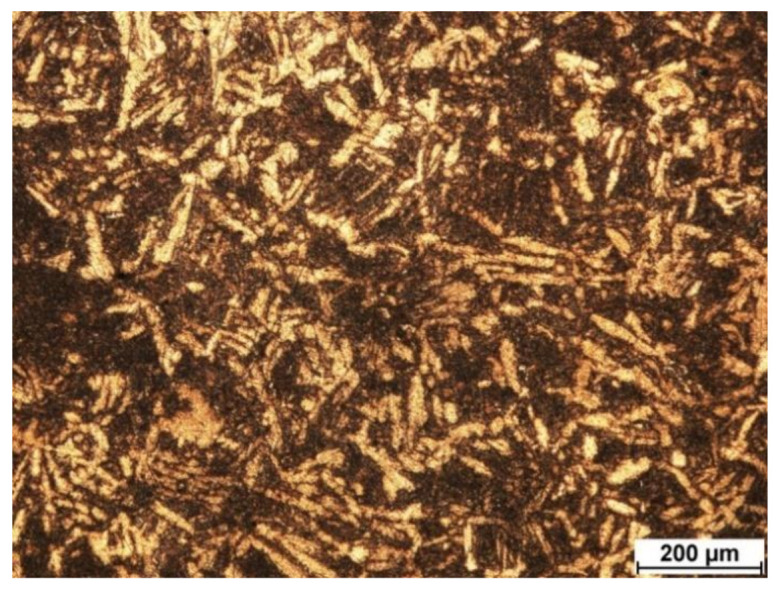
Microstructure of zirconium spark plasma sintered at 1400 °C and cooling rate of 400 °C/min.

**Figure 5 materials-14-05006-f005:**
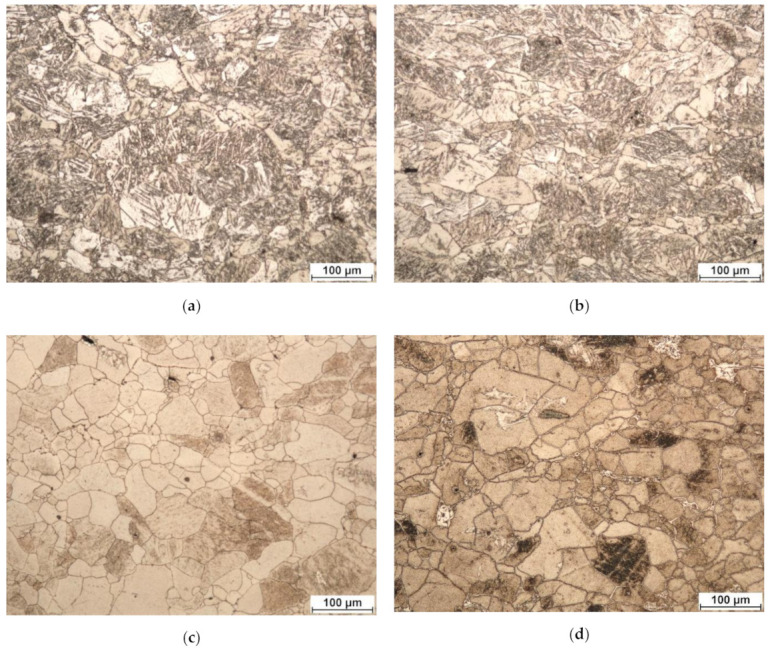
Microstructure of zirconium spark plasma sintered at 800 °C with various cooling rates and subsequent heat treatment at: (**a**) 400 °C/min; (**b**) 10 °C/min; (**c**) 400 °C/min + heat treatment at 580 °C and holding at 120 min; (**d**) 400 °C/min + heat treatment at 750 °C and holding time at 30 min.

**Figure 6 materials-14-05006-f006:**
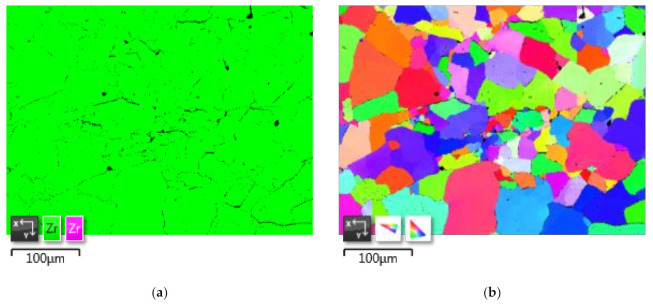
EBSD micrographs of zirconium spark plasma sintered at 800 °C and subsequent heat treatment at 580 °C with holding at 120 min: (**a**) phase map; (**b**) IPF map.

**Figure 7 materials-14-05006-f007:**
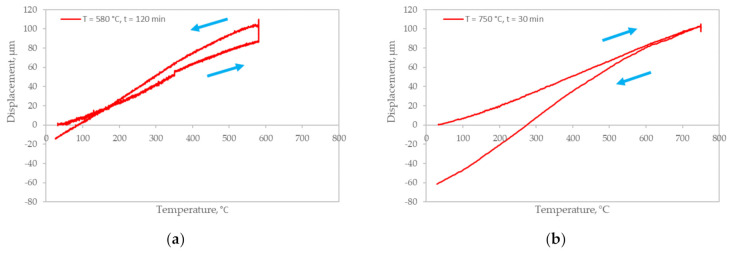
Dilatometric curves of spark plasma sintered zirconium at 800 °C with subsequent heat treatment at temperature and holding time: (**a**) 580 °C and 120 min; (**b**) 750 °C and 30 min.

**Figure 8 materials-14-05006-f008:**
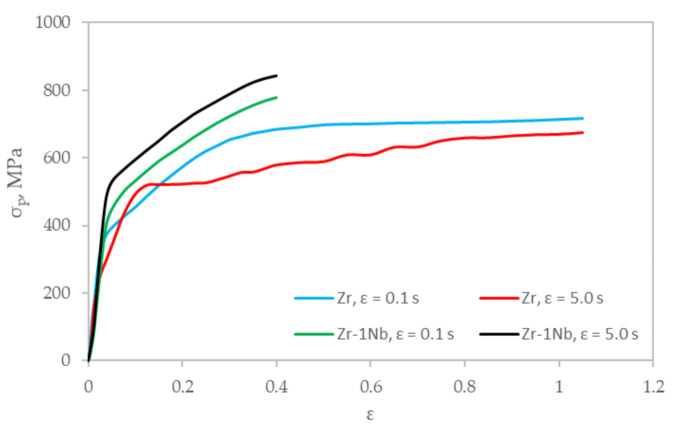
Influence of strain rate and strain level on change in deformation resistance (uniaxial compression method on Gleeble 3800 plastometer) of specimens Ø10 × 12 mm manufacture from Zr-1Nb alloy (traditional route) and zirconium (spark plasma sintering).

**Table 1 materials-14-05006-t001:** Chemical composition of zirconium powder.

Zr wt %	B wt %	Al ppm	Fe ppm	Si ppm	Mn ppm	Ni ppm	Ti ppm	Cr ppm	Hf ppm	O ppm	C ppm	N ppm	F ppm
99.80	0.43	30	163	<30	3	19	<30	18	361	730	36	29	180

**Table 2 materials-14-05006-t002:** Post-SPS heat treatment parameters.

SPS Temperature	T = 800 °C			
SPS cooling	Rapid	Rapid	Rapid	Slow
	400 °C/min	400 °C/min	400 °C/min	10 °C/min
Heat treatment temperature and time (residual pressure in vacuum no more than 1 × 10^−4^ torr)	$T=580_{-15}^{+10}\;{}^{\circ}C$	$T=750_{-20}^{+30}\;{}^{\circ}C$	–	–
	$t=120^{+20}\;\min$	$t=30^{+20}\;\min$		
Heat treatment heating & cooling	100 °C 10 °C until 300 °C then free cooling	100 °C 10 °C until 300 °C then free cooling	–	–

**Table 3 materials-14-05006-t003:** Bulk density and microhardness of zirconium spark plasma sintered at various temperatures and cooling rates.

Sintering Temperature, °C	Cooling Rate, °C/min	Bulk Density, g/cm^3^	Microhardness, HV_0.05_
700	400	6.38	135 ± 12
	10	6.39	139 ± 9
750	400	6.38	136 ± 15
	10	6.39	134 ± 14
800	400	6.40	151 ± 18
	10	6.44	151 ± 14
1400	400	6.48	150 ± 20

**Table 4 materials-14-05006-t004:** Comparison of strength properties of zirconium R60001 grade and zirconium spark plasma sintered at 800 °C meeting requirements of ASTM B-351 standard.

Sample	Yield Strength, MPa	Ultimate Tensile Strength, MPa
Zr (grade R60001)	>140	>290
Zr (SPS)	220	380

## Data Availability

All the data generated during this research are included in this published article.

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
