# Peer review of "Development of Alternative Method for Manufacturing Structural Zirconium Elements for Nuclear Engineering"

_materials, 2021, doi:10.3390/ma14175006_

Round 1

Reviewer 1 Report

Dear Authors,
I have read your paper "Development of an alternative method for manufacturing structural zirconium elements for nuclear engineering" carefully.
This paper describes the operation properties of the Zr material which were fabricated by mean of high concentrated energy treatments.
The paper is easy to read.
But the methods are not properly described, so that other research groups may not reproduce them.
The paper is interesting. However, it requires few corrections.
1. The introduction has no information about another paper about SPS of the Zr alloys, for example, https://doi.org/10.3390/ma14123172. Add more new references related to the field (2021, 2020)
2. Please, add information about the time of the indentation during the microhardness test.
3. Please, compare the microhardness with date https://doi.org/10.3390/ma14123172 (line 244)
4. A sketch about the SPS process and its steps would be useful. Did cold pressure of the powders before SPS?
5.  English is not the native language of this Reviewer, but proofreading is strongly recommended.
6. A question just for curiosity, normally the pipe will be manufactured through an SPS process?
The paper can be accepted for publication after minor improvements.

Reviewer 2 Report

The paper “Development of an alternative method for manufacturing structural zirconium elements for nuclear engineering” presented the preparation and characterization of zirconium parts prepared by Spark Plasma Sintering method (SPS). The mechanical tests showed promising results. Several critical issues need to be addressed before publication.

  1. As the SPS method was used as an alternative approach, it will be better to perform and summarize quantitative comparison for zirconium parts prepared by SPS and other traditional methods.

  1. Are there any potential issues of using the SPS method in preparing zirconium?

  1. The formats and English writing can be improved.

Reviewer 3 Report

Dear author,

The article have 2 main problems – confusion with EBSD phases and mechanical properties description and presentation, which shall be the main results. There is missing discussion – what shall be the results of experiment and what finally was the results.  

Table 2 is unclear.

P3 l.106                … were performed…

Figure 3 – x-axes shall be the same, otherwise the message is misleading

P6 l.162               mention measured property – hardness – in the sentence

P7 last paragraph            EBSD identify the crystal structure, whereas XRF or EDS composition. The sentence shall be improved.

Table 4 is superfluous. I would like to encourage author to write some points are not evaluated, it is realistic and serious from scientific point of view, but other parameters are or unknown to non-users of Oxford Instruments tool (as Phase Count) or it is unclear, what they really mean (Mean band contrast – 191703 good, but minor 81.77 –still acceptable, or not?) The terms shall be explained or removed.

Main problem of EBSD is that Zirconia refers to ZrO2, not to metal –zirconium. The majority of the surface is evaluated by cubic zirconia, but zirconium is hexagonal!!! Is it just misleading or completely false data?

Compare to quite nothing giving 3.3 part, the part 3.4 is very short. The most interesting results – deformation 500 % per second deformation curve, tensile strength – are not given. If tensile strength is 290 MPa, than it mean sample broke practically without any plasticity compare to visual yield strength 500 MPa from Fig. 8. Is really strain/deformation of SPS Zr over 100 %?

There is mentioned hP the height of sample when cracks are formed, but any cracks are not mentioned in discussion. Simply, there is missing too much important detail, whereas too much unnecessary details is mentioned.

There must be mentioned results of each experimental sets. Each important results must be presented and supported by curves and figures (fractography).

The SPS compactization of Zr is not new, there are published results from Kraków, Poland. The introduction must mention already published literature, even in English.

The article needs complete rework and resubmission. It have scientific soundness, but needs supervision and weighting of important and unimportant parts.

Reviewer 4 Report

This paper contain the meaningful experimental results about an alternative method for manufacturing structural zirconium.

Please consider the revision about the following points.

Specific comments.

Does it mean the displacement in Figure3? Please describe in the text.

Table2 Density unit g/cm3 → g/cm3

L180 10-4 → 10-4

L231 Figure5(c) →Figure 6(b)?

L240 150+10HV L241 160±10HV Is it correct?

L248 Table2? Is it correct?

L269 Is the formula correct?

What is the correlation between DIL changes in Figure 7 and displacement in Figure 3? Please describe in the text if there is a relationship.

L294, 322, 325, 329 S-1 → S-1

It will be easier to understand if there are color coding of â–² and â–  in Figure 8. What is the effect of the change of hierarchy around 0.1in Figure 8? Please describe it in the text.

L331 Please confirm the formula.

Round 2

Reviewer 2 Report

The revised manuscript is ready for publication. 

Reviewer 3 Report

Dear authors,

I would like to thank you for clearing uncertainties, mainly checking EBSD results.

Please, try to improve quality / resolution of both parts of Figure 6. It seems blurred and can be improved in graphical editor.

In Ref 17 there tags from previous html version.